# *Escherichia coli* Isolated from Diabetic Foot Osteomyelitis: Clonal Diversity, Resistance Profile, Virulence Potential, and Genome Adaptation

**DOI:** 10.3390/microorganisms9020380

**Published:** 2021-02-13

**Authors:** Alexi Lienard, Michel Hosny, Joanne Jneid, Sophie Schuldiner, Nicolas Cellier, Albert Sotto, Bernard La Scola, Jean-Philippe Lavigne, Alix Pantel

**Affiliations:** 1VBIC, INSERM U1047, Université de Montpellier, UFR de Médecine, 30908 Nîmes CEDEX 2, France; ALIENARD@ch-bassindethau.fr; 2Aix-Marseille Université UM63, Institut de Recherche pour le Développement IRD 198, Assistance Publique Hôpitaux de Marseille (AP-HM), Microbes, Evolution, Phylogeny and Infection (MEΦI), Institut Hospitalo-Universitaire (IHU) Méditerranée Infection, 13005 Marseille, France; michosny@hotmail.com (M.H.); joannejneid@hotmail.com (J.J.); bernard.la-scola@univ-amu.fr (B.L.S.); 3VBIC, INSERM U1047, Université de Montpellier, Service des Maladies Métaboliques et Endocriniennes, CHU Nîmes, 30029 Nîmes CEDEX 09, France; sophie.schuldiner@chu-nimes.fr; 4Service d’Orthopédie, CHU Nîmes, 30029 Nîmes CEDEX 09, France; nicolas.cellier@chu-nimes.fr; 5VBIC, INSERM U1047, Université de Montpellier, Service des Maladies Infectieuses et Tropicales, CHU Nîmes, 30029 Nîmes CEDEX 09, France; albert.sotto@chu-nimes.fr; 6VBIC, INSERM U1047, Université de Montpellier, Service de Microbiologie et Hygiène Hospitalière, CHU Nîmes, 30029 Nîmes CEDEX 09, France; alix.pantel@chu-nimes.fr

**Keywords:** adaptation, diabetic foot osteomyelitis, *Escherichia coli*, resistance, whole-genome sequencing, virulome

## Abstract

This study assessed the clonal diversity, the resistance profile and the virulence potential of *Escherichia coli* strains isolated from diabetic foot infection (DFI) and diabetic foot osteomyelitis (DFOM). A retrospective single-centre study was conducted on patients diagnosed with *E. coli* isolated from deep DFI and DFOM at Clinique du Pied Diabétique Gard-Occitanie (France) over a two-year period. Phylogenetic backgrounds, virulence factors (VFs) and antibiotic resistance profiles were determined. Whole-genome analysis of *E. coli* strains isolated from same patients at different periods were performed. From the two-years study period, 35 *E. coli* strains isolated from 33 patients were analysed; 73% were isolated from DFOM. The majority of the strains belonged to the virulent B2 and D phylogenetic groups (82%). These isolates exhibited a significant higher average of VFs number than strains belonging to other groups (*p* < 0.001). *papG2* gene was significantly more detected in strains belonging to B2 phylogroup isolated from DFI compared to DFOM (*p* = 0.003). The most prevalent antibiotic resistance pattern was observed for ampicillin (82%), cotrimoxazole (45%), and ciprofloxacin (33%). The genome analysis of strains isolated at two periods in DFOM showed a decrease of the genome size, and this decrease was more important for the strain isolated at nine months (vs. four months). A shared mutation on the putative acyl-CoA dehydrogenase-encoding gene *aidB* was observed on both strains. *E. coli* isolates from DFOM were highly genetically diverse with different pathogenicity traits. Their adaptation in the bone structure could require genome reduction and some important modifications in the balance virulence/resistance of the bacteria.

## 1. Introduction

Diabetic foot ulcers (DFUs) are estimated to affect 19 to 34% of all diabetic individuals during their lifetime [1]. Infection of these ulcers is frequent (40–80%), representing a major cause of mortality and morbidity [2]. Diabetic Foot Osteomyelitis (DFOM) is a common complication of DFU and/or diabetic foot infections (DFI) [3]. In Western developed countries, DFI are mainly caused by aerobic Gram-positive cocci (especially *Staphylococcus aureus*). However, in deep ulcers or wounds occurring in a patient who has been previously treated by antibiotics, DFI/DFOM are more often polymicrobial, including aerobic Gram-negative and obligate anaerobic bacteria. Furthermore, recent epidemiological studies from subtropical countries described a considerably higher predominance of Gram-negative bacilli (*Pseudomonas aeruginosa* and *Enterobacteriaceae*) in these geographical parts of the world [4].

The characterization of *Escherichia coli* isolates from skin and soft tissue infection (SSTIs) have been previously published [5,6]. However, this description still remains poor from DFIs, whereas *E. coli* is a main pathogenic Gram-negative bacteria isolated from these ulcers [7,8,9,10,11]. This pathogen is one of the most important agents of extraintestinal infections, with the potential to cause infections in almost any anatomical site. Extraintestinal Pathogenic *E. coli* (ExPEC) strains possess virulence factors (VFs) encoding genes that cause infections. These genes are located on plasmids or, more frequently, on the chromosome. On the chromosome, they are typically found in a specific region called Pathogenicity Island (PAI). Such VFs allow *E. coli* to bind to human cells (P-fimbriae, S-fimbriae), to survive in the human body (siderophores) and to invade the host by damaging human cells and tissues (toxins such as hemolysin and cytotoxic necrotizing factors) [12]. This combination of VFs determines if *E. coli* can cause infection [13]. The host characteristics also play an important role. Studies have described that the host compromised, including the older age and urinary tract abnormalities, allows comparatively low virulence *E. coli* strains to cause urosepsis [14]. 

In this study, we described *E. coli* strains isolated from DFOM/deep DFI, determined their phylogenetic relationships, virulence profiles, antibiotic resistance, and genome adaptation for two persistent infections.

## 2. Materials and Methods

### 2.1. Study Design

This study was approved by the institutional review board (IRB-2017-04) and carried out in accordance with the Helsinki Declaration as revised in 2000 [15]. From the 1 January 2015 to the 31 December 2016, we retrospectively enrolled all diabetic patients managed in the Clinique de Pied Diabétique Gard Occitanie at the Nîmes University Hospital (France) for deep DFI and suspected DFOM. Patients were included if they had not received any antibiotic agents in the previous week. All wounds were assessed for presence and severity of infection by a trained diabetologist using the PEDIS classification of the IWGDF consensus conference [2]. Patients were suspected of having osteomyelitis of the foot if they had at least two of the following clinical criteria: (i) a wound whose duration was ≥2 weeks located above an underlying bony prominence, with an area >2 cm^2^ or a depth >3 mm, (ii) a positive probe-to-bone test and (iii) abnormalities consistent with the diagnosis of osteomyelitis either on plain X-rays, radionuclide procedures (three-phase bone scan and/or labelled leukocyte imaging), or magnetic resonance imaging. DFOM was definitively diagnosed when one or more bacteria was isolated from bone biopsies. Epidemiological and clinical data were gathered for all patients.

After wound debridement, samples for bacterial culture were obtained by transcutaneous bone or tissue biopsy performed by a trained orthopaedist using the procedure previously described [3]. All the samples were immediately sent to the Department of Microbiology.

### 2.2. Conventional Microbiological Method 

Tissue and bone samples were cultured following the European guidelines [16]. Additionally, a Schaedler broth was inoculated by bone biopsy. All the media were incubated for 14 days. Genus and species of all the isolates were determined using the Vitek^®^ MS system (bioMérieux, Marcy-l’Etoile, France). Susceptibility to antimicrobial agents was tested by the disk diffusion method (BioRad, Marnes La Coquette, France) on Mueller-Hinton agar with or without horse blood, according to the recommendations of the EUCAST-SFM 2019 (http://www.sfm-microbiologie.org (28 January 2021)). In addition, Minimum Inhibitory Concentrations (MICs) of carbapenems (ertapenem, imipenem, and meropenem) were determined by E-test method (bioMérieux). MIC of colistin was determined using microbroth dilution (Umic^®^, Biocentric, France). The MICs were interpreted as specified by the CA-SFM/EUCAST criteria. *E. coli* resistant to the third generation cephalosporins were classified as multidrug-resistant organism (MDRO). 

### 2.3. Analysis of Clonality of the Strains 

The genetic relationship between the *E. coli* strains was evaluated by repetitive sequence-based PCR (rep-PCR) using the DiversiLab^®^ strain typing system (bioMérieux). Results were interpreted with DiversiLab web-based software (bioMérieux) using the Pearson correlation and the modified Kullback–Leibler method. Isolates with identical strain patterns were considered indistinguishable if the similarity percentage was ≥95%. 

Multi-Locus Sequence Typing (MLST) analysis was performed using the Achtman MLST scheme (pubmlst/org/mlst). Seven housekeeping genes (*edk, fumH, gyrB, icd, mdh, recA and purA*) were amplified according to this protocol. 

### 2.4. Phylogenetic Grouping 

Phylogenetic grouping of the *E. coli* strains was determined by a PCR-based method developed by Clermont et al. [17] identifying one of the eight phylogenetic groups (A, B1, B2, C, D, E, F, clade I) using a combination of four DNA markers (*chuA*, *yjaA*, *arpA* genes, and TspE4.C2).

### 2.5. Molecular Characterization of Main Resistance Genes 

Total DNA of cultures was extracted using the EZ1 DNA Tissue kit on the BioRobot EZ1 extraction platform (Qiagen, Courtaboeuf, France). Genes encoding the most clinically prevalent Extended Spectrum β-Lactamases (ESBLs) (*bla*_TEM_, *bla*_SHV_, and *bla*_CTX-M_) were detected by PCR using specific primers and confirmed by sequencing the PCR products, as described previously [18,19]. A multiplex PCR was used for the detection of plasmidic *bla_ampC_* genes [20]. Genetic characterization of the quinolone resistance-determining region (QRDR) (*gyrA*, *gyrB*, *parC* and *parE* mutations) was performed by PCR and gene sequencing [21].

### 2.6. Virulence Genotyping 

The *E. coli* strains were tested by PCR for the presence of a panel of 20 genes encoding known VFs [21,22,23,24,25]: 1) Fimbriae and/or adhesins: *fimH* (D-mannose-specific adhesin, type 1 fimbriae), *papG1, papG2, papG3* (Gal(α1-4)Gal-specific pilus tip adhesin molecule), *papA* (major structural subunit of P fimbrial shaft; defines F antigen), *papC* (pilus assembly; central region of pap operon), *papE, sfaS*, *focG* (S fimbriae and F1C fimbriae), *afa/draBC* (Dr family adhesin); 2) Toxins: *cnf1* (cytotoxic necrotizing factor-1), *hlyA* (hemolysin); 3) Iron uptake: *iutA* (ferric aerobactin receptor (iron uptake: transport), *iroN* (catecholate (salmochelin) siderophore receptor), *fyuA* (ferric yersiniabactin uptake receptor); 4) Protectins: *kpsMT*II (capsule synthesis), *traT* (surface exclusion, serum survival (outer membrane protein)); 5) Others: *usp* (uropathogenic-specific protein (bacteriocin)), *malX* (a marker for pathogenicity-associated island marker from archetypal uropathogenic strain CFT073), and *ompT* (outer membrane protein (protease) T).

If a strain was positive for two or more markers including *papAH* and/or *papC*, *sfa/focDE*, *afa/draBC*, *kpsMTII*, and *iutA*, this isolate belonged to ExPEC [26].

### 2.7. Whole-Genome Analysis and Single-Nucleotide Polymorphism Identification 

*E. coli* strains (*n* = 4) isolated from same patients at different periods were sequenced. Whole Genome Sequencing (WGS) was performed with an Illumina MiSeq sequencing system (Illumina, San Diego, CA, USA) using paired-end (PE) read libraries (PE250) prepared by Nextera XT DNA Library Prep Kit (Illumina) following the manufacturer’s protocol. Rax reads were processed using FastQC (v.0.11.7) to assess data quality. The Cutadapter tool (v.1.16) implemented in Python (v.3.5.2) was used to remove residual PCR primers and to filter low quality bases (Q_score <30) and short reads (<150 bp). The filtered trimmed reads were included in the downstream analysis. Obtained reads were mapped against *E. coli* IAI39 genome (GenBank accession number: GCA_000026345.1), using the CLC genomics workbench 7 (Qiagen Inc., Valencia, CA, USA). The assembled contigs were processed by Prokka software for microbial genome annotation [27]. The VirulenceFinder-1.5 server (https://cge.cbs.dtu.dk/services/VirulenceFinder (28 January 2021)) was used to infer virulence encoding genes from genomes sequences. Antimicrobial resistance genes and plasmid replicons were obtained from ABRIcate with the ResFinder database on assembled genomes [28,29] and PlasmidFinder database [30]. Toxin-antitoxin encoding ORF were extracted from genomes annotations. Targeted genomes were aligned binary against wild-type strains using the MAFFT software, as following NECS21/NECS50 and NECR70/NECR107 [31]. SNP calls were made from the PE library raw reads. The REALPHY tool was used to identify the closest relationships between the strains and the different *E. coli* reference genomes deposited in GenBank. For SNP analysis, we employed the following software: SNP-sites for variants calling [32] and SnpEff (v.4.3T) for SNP annotation in coding regions [33]. SNP annotations of affected genes were searched within wild-type genomes and their effects were classified depending mutations impacts. Genes affected by stop gain mutations were searched for in the Uniprot database for virulence classification. Finally, PAIs were predicted within implicated genomes using IslandViewer 4 database [34].

### 2.8. Statistical Analysis 

The analyses were merely descriptive: data were given as numbers and percentages. For each VF, comparisons between the different phylogroups were evaluated by using Fisher’s exact test. A *p-*value ≤ 0.05 was considered as reflecting statistical significance. Statistical analyses were performed using GraphPad Prism 6.0 (GraphPad Software, La Jolla, CA, USA).

## 3. Results

### 3.1. Baseline Characteristics and Microbiological Considerations 

From the two-year study period, 340 patients were hospitalized for DFI and included into the study. A total of 704 strains were isolated (Appendix A) from 362 samples, corresponding to a mean number of 1.94 isolate per sample. A polymicrobial infection was present in 206 samples (7 with 5 bacteria, 45 with 4 bacteria, 77 with 3 bacteria, 77 with 2 bacteria). In 104 samples, infection was monomicrobial and 52 cultures were negative. Aerobic Gram-positive cocci were predominant (46.5% of all species) with *S. aureus* as the most commonly isolated pathogen (*n* = 162) accounting for 23.0% of the species (44.5% of the Gram-positive cocci). *Streptococcus* spp. represented 11.1% of all species (*n* = 78). Aerobic Gram-negative bacilli accounted for 28.8% of the microorganisms (*n* = 203). Among them, *Enterobacteriaceae* were the most frequent bacteria (25.7% of all species) especially *E. coli,* which represented 19.9% of enterobacteria. Non-fermentative Gram-negative bacilli were rather uncommon with notably *Pseudomonas* spp. representing exclusively 1% of all species. Finally, the anaerobes represented 19.0% of the species. *Finegoldia* spp. and *Bacteroides* spp. were the most commonly isolated anaerobes (14.9% of all species). 

As previously mentioned, *E. coli* was the main enterobacteria detected (*n* = 36) and were isolated in 34 patients from the bacterial culture of their infected wound. One strain was not investigated because no subculture could be obtained after congelation. We included 33 patients. Four samples (12%) were monomicrobial, 14 (42%) were bi-microbial, the others (15, 46%) contained three or more microorganisms. 

The demographic and clinical characteristics of the 33 patients definitively included are shown in Table 1. 

Most of the included patients were male (82%) with a median age of 66.5 years (36–96). Thirty-two DFI (97.1%) were classified as grade 3 and one as grade 4 (2.9%). In 16 patients (48.5%), the current wound was the first episode of ulceration. The majority of the wounds (73%) were associated with osteomyelitis. Two patients had a persistent infection with two E. coli isolated at 4 months (NECS21-NECS50) and 9 months apart (NECR70-NECR107). 

### 3.2. Antimicrobial Susceptibility of E. coli 

The in vitro activities of antimicrobial agents against the 33 *E. coli* isolates are presented in Table 2. Three strains (9%) expressed an extended spectrum β-lactamase (ESBL), which confers resistance to third generation cephalosporins.

ESBL were carried by *bla*_CTX-M-1_ gene (associated with a *bla*_TEM-1_ gene) for two strains, and by *bla*_CTX-M-9_ gene for one strain. Four strains (12%) were resistant to piperacillin/tazobactam. All studied isolates were susceptible to ertapenem and imipenem. Amikacin was the most effective aminoglycoside (only 1 resistant strain). The most prevalent antimicrobial resistance was observed for ampicillin (82% of the strains), cotrimoxazole (45%) and amoxicillin/clavulanic acid (42%). Resistance to ciprofloxacin was found in 11 strains (33%) due to mutations in *gyrA* (at codon 83 (Ser 83→ Tyr) and/or codon 87 (Asp87→Asn) and *parC* genes (at codon 80 (Ser 80→Ile) and/or codon 84 (Glu84→Ala)).

### 3.3. Clonality Analysis of the E. coli 

The rep-PCR showed a great diversity of the *E. coli* while the strains were grouped in 27 different rep-PCR patterns (Figure 1). Only two profiles included three isolates: patterns IX (*n* = 3; phylogroup B2), XIX (*n* = 3; B2). The residual patterns contained two isolates (4 rep-PCR patterns: VIII, XXIII, XXV, XXVII) or single isolates (21 rep-PCR patterns). Among the isolates harboring the same rep-PCR pattern, we observed that the strains isolated at two periods in two patients were identical in each case (patterns XXIII and XXVII). 

The Achtman MLST scheme also showed the great diversity of the isolates with 18 ST detected. The ST95 was the most important (6 isolates; 3 concerning pattern XIX) followed by ST597 (5 isolates; including pattern XXIII), ST73 (4 isolates and including pattern IX), and ST31 (4 isolates, including pattern XXVII). No isolate belonged to the international O25b:H4-ST131 clone.

### 3.4. Virulence Profiles of E. coli 

The distribution of phylogroups and virulence factors are presented in Table 3. 

The most prevalent phylogenetic group was B2 (54%, *n* = 19) followed by D phylotype (28%, *n* = 10), B1 (9%, *n* = 3), C (6%, *n* = 2), and clade I (3%, *n* = 1). The majority of the strains (71%) could be defined as ExPEC according to the definition [26]. The average number of VFs was 10.1 (±3.7). Strains belonging to group B2 exhibited a significant higher average of VFs number than strains belonging to B1 (11.8 *vs* 5.0, *p* < 0.001). 

NS, not significant.Concerning the genes encoding the adhesins, all the strains harboured *fimH*. Half of the strains presented *papA*, *papC, papE* genes (17 isolates, 49%). *papG2* was most frequently detected compared to *papG3* (14 (40%) vs. 2 (5.7%) isolates, respectively). No strain harboured the *papG1* gene. Concerning the genes encoding toxins, *hlyA* and *cnf1* genes were detected in 12 (35%) and 10 isolates (28%), respectively. The majority of isolates presented the iron uptake and protectins encoding genes: *fyuA* (*n* = 31, 88%), *kpsMT*II (*n* = 24, 69%), *traT* (*n* = 23, 66%), *iroN* (*n* = 20, 57%) and *iutA* (*n* = 20, 57%). Finally, the protease-encoding gene (*ompT*) was detected in almost all the strains (*n* = 32, 91%). As we mentioned below, the strains belonging to B2 phylogroup possessed the majority of the VFs screened. Interestingly, *afa1* (*n* = 1, 5% of the strains belonging to B2), *papG3* (*n* = 2, 11%), *cnf1* (*n* = 10, 53%), *hlyA* (*n* = 12, 63%), and *usp* (*n* = 19, 100%) genes were exclusively detected in this group of strains. Moreover, all of these strains harboured *fimH*, *usp,* and *malX* genes. 

The distribution of strains isolated from DFOM was similar of that of the deep foot infections suggesting no particular tropism of bone for a clone: 46% belonged to the phylogroup B2, 34% to the D, 8% to B1 and C, 4% to clade I. Strains involved in DFOM exhibited less VFs than strains involved in deep DFI (9.5 vs. 12.1, respectively) even if this difference was not significant (Table 3). Concerning the distribution of VFs, we could note that *papG3* gene were exclusively detected in DFOM and the isolates found in bone biopsies presented more *irp2* and *fyuA* genes (Table 3). On the other side, *papG2* gene was significantly more detected in strains belonging to B2 phylogroup isolated from deep foot infections compared to DFOM (*p* = 0.003). A trend was also noted for *sfaS* and *traT* genes more frequently present in deep DFI (*p* = 0.07 and 0.06, respectively). 

Finally, we noticed that the most resistant strain (harbouring ESBL, resistance to aminoglycosides, fluroquinolones, and cotrimoxazole) was also the strain with the lower number of VFs and belonged to the B1 phylogroup. The other ESBL-producing isolates belonged to B2 phylogroups and harboured an important arsenal of VFs (8 and 14, respectively). These isolates were equally distributed between DFOM and deep DFI (2 vs. 1, respectively). The most susceptible strains to antibiotics more commonly belonged to the B2 phylogroup (6 strains/6). 

### 3.5. E. coli Genome Analysis 

In this study, two patients with DFOM harboured *E. coli* at two time periods. Following the result of rep-PCR indicating that these isolates were clonal, we sequenced the genomes of the strains (Table 4). 

We confirmed that the strains were identical in each case for the first and the second sample. Moreover, virulence encoding genes and toxin/antitoxin ORFs were identified in all isolates (Appendix A). The average length of the mapped genomes studied was 4,473,412 base pair (bp). The greatest genome size was 4,528,137 bp (strain NECR70), the smallest genome size was 4,440,985 bp (strain NECR107).

Interestingly, in the two cases, the genome sizes were decreased in the second sample and this decrease was more important for the strain isolated at 9 months (vs. 4 months) (–87,152 vs. –13,374, respectively) (Table 4). An average of 4280 open-reading frames (ORFs) were predicted. 

Variant call analysis showed the presence of SNPs within coding regions, comparing *E. coli* isolates against wild-type strains (Appendix A). When evaluating SNPs in NECS50 strain, 1764 nucleotides were identified, and affected 30% of genes (1277/4284), as far as in the case of NECR107 strain, 6874 positions were annotated and dispersed within 51.5% of ORFs (2198/4267). Only Stop gain mutations were implicated in this analysis and affected genes were analysed depending on its function (virulence factors and other gene). Virulence encoding and non-virulence encoding genes were affected. Putative acyl-CoA dehydrogenase-encoding gene *aidB* was the unique gene shared by both strains (results are summarized in Table 5). PAIs were identified within all analysed genomes. A loss of some parts of these regions were noticed when comparing both type NECR107 and NECS50 strains with their wild-type strains (NECR70 and NECS21, respectively) (Figure 2). 

Only *multidrug transporter A* gene (*mdf(A)*) was identified in the four strains, with a coverage of 100% for all isolates, and an identity varying between 97.57% (NECR70, NECR107) to 98.14% (NECS21, NECS50). *mdf(A)* has a broad-spectrum specificity that include erythromycin, tetracycline, rifampicin, kanamycin, chloramphenicol, and ciprofloxacin. Neither other resistant markers, nor plasmids, were detected.

## 4. Discussion

This study characterised the *E. coli* strains isolated from DFOM and deep DFI and demonstrated that these bacteria were highly genetically diverse with different pathogenicity traits. Among them, some strains belonged to ST73, a major clonal complex isolated from UTI particularly pyelonephritis and associated with multidrug resistance profile [35,36,37]. Moreover, these strains were isolated from monomicrobial infection in our panel confirming their virulence potential. 

As previously observed for SSTI, the studied strains belonged mainly to ExPEC strains, harboring a higher prevalence of seven known virulence factors. These virulence factors comprised adhesins, iron acquisition systems (e.g., aerobactin synthesis and uptake), and host defense avoidance mechanisms (e.g., cytotoxins, capsule) [38]. This study also highlighted that *E. coli* from diverse origins (with different ST) were capable of causing similar infections. They presented patterns of virulence, genomic and functional properties important in the pathophysiology, even if they were not specifically associated with a pathotype [6]. As also previously observed for *E. coli* isolated from SSTI [5] and osteomyelitis [39], the majority of the strains belonged to the B2 and D phylogroups, typically associated with more virulent strains [5,6,38]. However, it is interesting to note that some commensal *E. coli* (*n* = 6, 17%) belonging to non-B2 and -D phylogroups and which are not ExPEC, can also be isolated in DFOM, demonstrating an environmental adaptation modifying the pathogenicity of the “opportunistic” pathogen and the influence of host immunosuppression. Globally, *E. coli* strains isolated from DFOM exhibited less VFs than strains involved in deep DFI (Table 3). However, *papG2*, *sfaS* and *traT* genes were more frequently present in B2 strains isolated from deep DFI whereas *papG3*, *irp2* and *fyuA* genes were more detected in DFOM. Adhesion is mainly due to type I fimbriae (*fimH*), present in all the strains. This characteristic of our population was not previously found in SSTI [6] but observed in OM [39] suggesting the importance of this adhesin to reach the bone. This is an important point due to the very low ability of *E. coli* to adhere to osteoblasts [39,40]. The presence of various adhesins could be the way to maintain this bacterium in the infrastructure of the bone, hijacking host defense. In this way, we observed that our strains harboured more *papG2* genes compared to other studies on SSTI and OM (40% vs. 10–20%) [5,6]. Thus, *fimH, papG2* and *sfa* could represent the different solutions developed by *E. coli* to establish infection in bone structure. Siderophore production is also important for bacterial survival [41]. Interestingly, the strains isolated from DFOM harboured a high prevalence of siderophore encoding genes, suggesting the importance to find iron for the bacterial survival and multiplication in this stringent environment (as suggested by [6]). This was associated with the prevalence of *cnf1* and *hlyA* (29% and 34%, respectively), at a level similar to those found among SSTI [5], OM [39] and UTI [42,43,44]. These two toxins-encoding genes were associated in 75% of the strains possessing it, because *cnf1* and *hlyA* (among the hemolysin operon *hlyCABD*) were present in the pathogenicity island II (PAI II) J96 [45,46]. The HlyA toxin is particularly important to involve host cell apoptosis or necrosis/lysis (e.g., erythrocytes) and facilitate nutrient acquisition and iron liberation [30]. Crémet et al. highlighted the cytolytic activity of the HlyA against MG-63 osteoblastic cells when it was associated with the other Cnf1 toxin contributing to explain the virulence of our panel strains [39]. Finally, our isolates harboured a high prevalence of *kpsMT*II, *traT*, and *ompT* compared to SSTI or OM previously published [5,6,39]. This highlights that these *E. coli* could produce an important arsenal of “weapons” to avoid host response. 

In this study, we determined for the first time the longitudinal evolution of *E. coli* genomes present at different times in bone. Only two patients (6% of our panel) presented this situation, suggesting that this event was rare in DFOM. In these two patients, the late isolate was derived from its early counterpart. For the first patient, *E. coli* NECR70 and NECR107 belonged to the ST 597, D phylogroup, and were found at nine-month intervals; for the second, *E. coli* NECS21 and NECS50 belonged to the ST31, D phylogroup, and were found at four-month intervals. The comparative genome analysis showed that the genome content of the strains after some months in bone decreased. This observation is the reverse of the classical genome expansion in *E. coli*, which is associated with the acquisition of PAIs and VFs by horizontal gene transfer contributing to the development of infections [47]. Here, the bacterial adaptation of a “bone life” involved some genetic rearrangements. The longer the period of this “bone life” was, the more the VFs encoding and non-encoding genes were affected (Table 5). Interestingly, the virulome, the resistome and the housekeeping genes were modified. Thus, we could note the absence of *hlyA* and *cnf1*, two genes affecting the virulence as seen above. This suggests that *E. coli* strains modify their virulence by generating an adapted microbial population in the aim to survive in the bone and limit the host immune response as described [48]. Among the two collection strains, only one common gene was mutated: the putative acyl-CoA dehydrogenase AidB-encoding gene. AidB is a protein expressed within the Ada response, related in sequence to the acyl-coenzyme A (acyl-CoA) dehydrogenase family (ACADs) [49,50]. The Ada response plays an important role in protecting cells against the cytotoxic and mutagenic action of alkylating agents. This Ada response involves the expression of four genes creating the Ada operon (*ada, alkA, alkB, aidB*) that encodes for four proteins (Ada, AlkA, AlkB, AidB) playing specialized functions in removing alkylating lesions from DNA and RNA [49]. An effect of *aidB* mutation has been observed at low, sublethal doses of alkylating agents, indicating an AidB role in DNA protection against by-products of cell metabolism during stationary phase [51,52,53]. As the isolated strain (NECR107) at nine months harbored more mutations than the strain (NECS50) isolated at four months (Table 5), we could hypothesize that the deregulation of the Ada operon could cause some cellular abnormalities limiting the repair function of *E. coli*. Thus, this is interesting to note that *E. coli* after 9 months had some genome mutations affecting the “machinery” of the bacteria (e.g., Alanine-tRNA ligase-, DNA polymerase-, ribonuclease-encoding genes, *tsaE, ftsk*), the membrane integrity (e.g., *glcA, yqiJ, yqiK*), and the bacterial virulence (affecting biofilm or stress resistance). Conversely, these mutations increased the bacterial resistance to antibiotics (e.g., *gyrB, ygiC*). Finally, we could observe genetic modifications on *ydeH, yddA*, or cytoplasmic alpha-amylase encoding gene. The mutations of these genes have been previously associated with bacterial persistence and adaptation to long term survival [54,55], a state clearly observed in our study. 

Finally, a study highlights the link between bacterial traits and host characteristics [14]. The authors showed that in urosepsis, the most virulent isolates were isolated from younger, urological intact women, whereas the less virulent were preferentially isolated from older, urologically compromised men [14]. Furthermore, it has been shown that the prevalence and severity of DFIs are a consequence of host-related processes (e.g., immunopathy, neuropathy, and arteriopathy) and pathogen-related factors (e.g., virulence, antibiotic-resistance, and microbial organization) [56,57]. We could suggest that this host characteristic of diabetic patients clearly influences the infection and even if some *E. coli* had low virulence traits, this could be enough to establish a deep infection. 

## 5. Conclusions

This study indicates that a majority of ExPEC are responsible of DFOM and deep DFI. The adaptation of *E. coli* under stressful conditions (in bone cells) could involve a genome reduction and some important modifications in the metabolism and the balance virulence/resistance of the bacteria. This may suggest to clinicians to optimize antibiotic therapy against *E. coli* found in DFI to avoid their implantation in the bone.

## Figures and Tables

**Figure 1 microorganisms-09-00380-f001:**
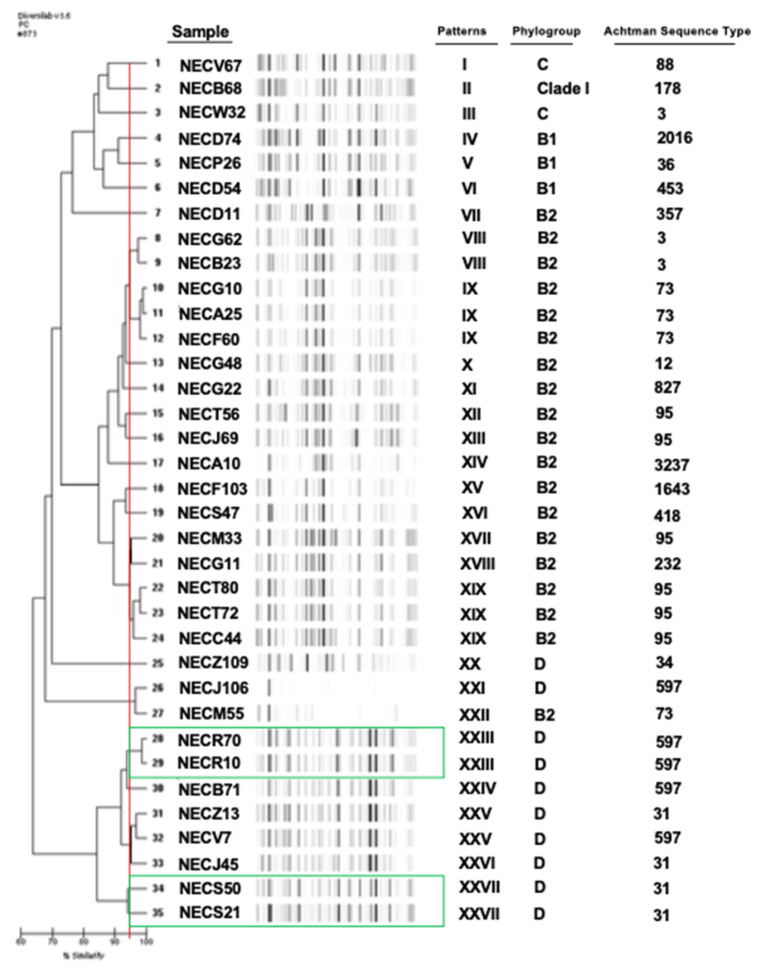
Genetic diversity of *Escherichia coli* strains isolated from diabetic foot infections using DiversiLab method, Multi-Locus Sequence Typing and phylogrouping. In green, the strains isolated at two periods in a same patient (Patient 1, NECR70/NECR10; Patient 2, NECS50/NECS21).

**Figure 2 microorganisms-09-00380-f002:**
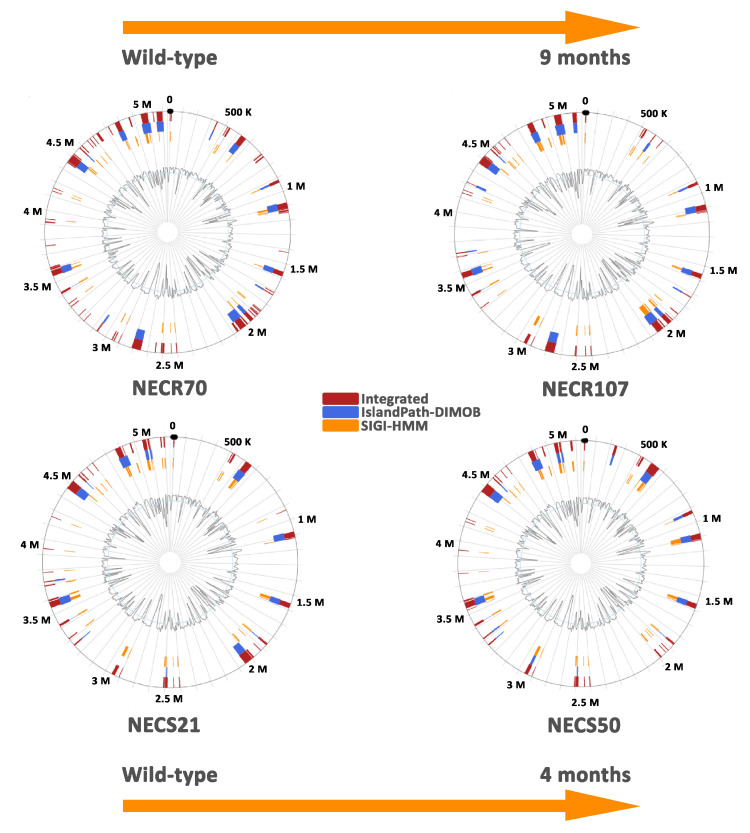
Prediction of pathogenicity islands within *Escherichia coli* genomes isolated from diabetic foot infections. Pathogenicity islands were predicted using IslandViewer 4, including three predictions methods Integrated, IslandPath-DIMOB, and SIGI-HMM (http://www.pathogenomics.sfu.ca/islandviewer (accessed on 28 January 2021)).

**Table 1 microorganisms-09-00380-t001:** Demographic and clinical characteristics of the studied patients.

Characteristics	Total *
Number of patients (total *n*)	33
Median Age (range), years	63.0 (36–96)
Male/female, *n* (%)	28 (84.8)/ 5 (15.2)
Type 1/type 2 diabetes mellitus	4 (12.1)/29 (87.9)
Mean diabetes duration (years)	16.7 ± 9.1
HbA_1c_ (%), mean	6.50 ± 0.89
Cardiovascular disease	
Absence	2 (6.7)
Coronary heart disease	10 (30.3)
Peripheral arterial disease	31 (93.9)
Neuropathy	33 (100)
Nephropathy	17 (51.5)
Diabetic retinopathy	14 (42.4)
Lifestyle factors	
Hypertension	17 (51.5)
Obesity	5 (15.1)
Dyslipidemia	18 (54.5)
First wound/Recurrence	14 (42.4)/19 (57.6)
Osteomyelitis	24 (72.7)
PEDIS infection classification	
Grade 3 (Moderate)	32 (97.0)
Grade 4 (severe)	1 (3.0)

* Values median and interquartile ranges (25th to 75th percentile) or numbers and percentages into brackets. HbA1c: glycated hemoglobin.

**Table 2 microorganisms-09-00380-t002:** Resistance of the 35 *E. coli* strains isolated from deep diabetic foot infections and diabetic foot osteomyelitis.

Antibiotics	Phylogroups, *n* (%)					
	B1	B2	C	D	I	Total
	3 (8.6)	19 (54.3)	2 (5.7)	10 (28.6)	1 (2.9)	35 (100)
AMX	3 (100)	13 (68)	2 (100)	9 (90)	1	28 (80)
AMC	2 (67)	10 (53)	1 (50)	3 (30)	0	16 (45.7)
TZP	1 (33)	2 (11)	0(0)	1 (10)	0	4 (12)
CTX	1 (33)	2 (11)	0 (0)	0 (0)	0	3 (8.6)
CAZ	1 (33)	2 (11)	0 (0)	0 (0)	0	3 (8.6)
FOX	0 (0)	0 (0)	0 (0)	0 (0)	0	0 (0)
IPM	0 (0)	0 (0)	0 (0)	0 (0)	0	0 (0)
OFX	3 (100)	3 (15.8)	1 (50)	4 (40)	1	12 (34.3)
CIP	3 (100)	3 (15.8)	1 (50)	4 (40)	1	12 (34.3)
GEN	1 (33)	0 (0)	0 (0)	0 (0)	0	1 (2.9)
TOB	1 (33)	0 (0)	0 (0)	0 (0)	0	1 (2.9)
AMK	0 (0)	0 (0)	0 (0)	0 (0)	0	0 (0)
FOS	0 (0)	0 (0)	0 (0)	0 (0)	0	0 (0)
SXT	2 (67)	4 (21)	1 (50)	3 (30)	1	11 (31.4)
COL	0 (0)	0 (0)	0 (0)	0 (0)	0	0 (0)

AMX, amoxicillin; AMC, amoxicillin + clavulanic acid; TZP, piperacillin/tazobactam; CTX, cefotaxime; CAZ, ceftazidime; FOX, cefoxitin; IPM, imipenem; OFX, ofloxacin; CIP, ciprofloxacin; GEN, gentamicin; TOB, tobramycin, AMK, amikacin; FOS, fosfomycin; SXT, cotrimoxazole; COL, colistin.

**Table 3 microorganisms-09-00380-t003:** Distribution of virulence factors in the *E. coli* isolated from diabetic foot osteomyelitis (DFOM) and deep diabetic foot infection (dDFI).

		DFOM				dDFI				Total	*p*	
		B2	D	Others	Total	B2	D	Others	Total		Total	B2
Number of strains		12 (46.2)	9 (34.6)	5 (19.2)	26 (74.3)	7 (78)	1	1	9 (25.7)	35 (100)	DFOM vs. dDFI	DFOM vs. dDFI
Mean number of VFs		10.8	8.7	7.2	9.5 ± 2.4	13.4	10	5	12.0 ± 3.5	10.2 ± 3.1	NS	NS
Adhesins	*papG* Class I	0 (0)	0 (0)	0 (0)	0 (0)	0 (0)	0	0	0 (0)	0 (0)	NS	NS
	Class II	1 (8)	6 (67)	2	9 (34.6)	4 (57)	1	0	5 (56)	14 (40)	NS	0.03
	Class III	2 (17)	0 (0)	0 (0)	2 (7.7)	0 (0)	0	0	0 (0)	2 (5.7)	NS	NS
	No	9 (75)	3 (33)	3	15 (57.7)	3 (43)	0	1	4 (44)	19 (54.3)	NS	NS
	*papA*	4 (33)	6 (67)	2	12 (46.2)	4 (57)	1	0	5 (56)	17 (48.6)	NS	NS
	*papC*	4 (33)	6 (67)	2	12 (46.2)	4 (57)	1	0	5 (56)	17 (48.6)	NS	NS
	*papE*	4 (33)	6 (67)	2	12 (46.2)	4 (57)	1	0	5 (56)	17 (48.6)	NS	NS
	*sfaS/focG*	3 (25)	0 (0)	1	4 (15.4)	5 (71)	0	0	5 (56)	9 (25.7)	NS	0.07
	*afa/draBC*	0 (0)	0 (0)	0 (0)	0 (0)	1 (14)	0	0	1 (11)	1 (2.9)	NS	NS
	*fimH*	12 (100)	9 (100)	5 (100)	26 (100)	7 (100)	1	1	9 (100)	35 (100)	NS	NS
Toxins	*hlyA*	8 (67)	0 (0)	0 (0)	8 (30.8)	4 (57)	0	0	4 (44)	12 (34.3)	NS	NS
	*cnf1*	7 (58)	0 (0)	0 (0)	7 (26.9)	3 (43)	0	0	3 (33)	10 (28.6)	NS	NS
Siderophores	*iutA*	4 (33)	7 (78)	3	14 (53.8)	5 (71)	1	0	6 (67)	20 (57.1)	NS	NS
	*irp2*	11 (92)	8 (89)	4	23 (88.5)	7 (100)	0	0	7 (78)	30 (85.7)	NS	NS
	*iroN*	8 (67)	2 (22)	2	12 (46.2)	6 (86)	1	1	8 (89)	20 (57.1)	NS	NS
	*fyuA*	12 (100)	8 (89)	4	24 (92.3)	7 (100)	0 (0)	0	7 (78)	31 (88.6)	NS	NS
Protectins	*kpsMTII*	9 (75)	6 (67)	1	16 (61.5)	7 (100)	1	0	8 (89)	24 (68.6)	NS	NS
	*traT*	4 (33)	7 (78)	4	15 (57.7)	6 (86)	1	1	8 (89)	23 (65.7)	NS	0.06
Miscellanous	*ompT*	12 (100)	8 (89)	4	24 (92.3)	6 (86)	1	1	8 (89)	32 (91.4)	NS	NS
	*malX*	12 (100)	3 (33)	1	16 (61.5)	7 (100)	0 (0)	1	8 (89)	24 (68.6)	NS	NS
	*usp*	12 (100)	0 (0)	0 (0)	12 (46.2)	7 (100)	0 (0)	0 (0)	7 (78)	19 (54.3)	NS	NS

**Table 4 microorganisms-09-00380-t004:** *Escherichia coli* genomic features isolated from DFOM of two patients at two periods.

Strain	Patients	Date of Isolation (Dy/Mo/Y)	Genome Coverage (fold)	Sequence Type	Genome Size (bp)	G+C Content (%)	Coding %	No. of ORFs ^a^
NECS21	57 y, DT2 ^b^	11/03/2015	101×	ST3	4,468,950	51.1%	87.4%	4284
NECS50		21/07/2015	151×		4,455,576	51.1%	86.5%	4258
NECR70	67 y, DT2	22/10/2015	105×	ST494	4,528,137	51.1%	87.2%	4312
NECR107		30/07/2016	98×		4,440,985	51.2%	86.4%	4267

^a^ Open-Reading Frame; ^b^ DT2, diabetes mellitus Type 2.

**Table 5 microorganisms-09-00380-t005:** Comparison between wild-type and target strains based on Stop gain affected genes.

NECS21/ NECS50	NECR70/ NECR107
**Genes non-encoding for virulence factors**	
Hypothetical protein (6×) Putative acyl-CoA dehydrogenase AidB * Branched-chain-amino-acid aminotransferase PTS system maltose-specific EIICB component Sensor histidine kinase YehU Putative lipoprotein YfhM Inner membrane protein YjiY Pyrimidine 5’-nucleotidase YjjG	Hypothetical protein (28×) Putative acyl-CoA dehydrogenase AidB * Alanine--tRNA ligase machinerie synthèse prot Guo M Nature 2009 Allantoinase Anguibactin system regulator 8-amino-7-oxononanoate synthase Multifunctional CCA protein DNA polymerase IV 1 Hexuronate transporter DNA translocase FtsK segregation chromosome Glycolate permease GlcA transporteur mb Malate synthase G Bifunctional glutamine synthetase adenylyltransferase/adenylyl-removing enzyme DNA gyrase subunit B Oxygen-independent coproporphyrinogen-III oxidase-like protein YqeR 5-hydroxyisourate hydrolase Putative defective protein IntQ Prophage integrase IntS Aerobactin synthase Group II intron-encoded protein LtrA Lactate utilization protein A System maltose-specific EIICB component M annosyl-D-glycerate transport/metabolism system MngR Transcriptional repressor MprA Protein-methionine-sulfoxide reductase catalytic subunit MsrP Protein-methionine-sulfoxide reductase heme-binding subunit MsrQ Adenine DNA glycosylase Na+/H+ antiporter NhaB Bifunctional NAD(P)H-hydrate repair enzyme Nnr Oligoribonuclease Putrescine aminotransferase Glucose-6-phosphate isomerase Putative glycerol-3-phosphate acyltransferase AD-dependent dihydropyrimidine dehydrogenase subunit PreA Epoxyqueuosine reductase dITP/XTP pyrophosphatase 3,4-dihydroxy-2-butanone 4-phosphate synthase 23S rRNA (guanosine-2’-O-)-methyltransferase RlmB Ribonuclease R 50S ribosomal protein L16 3-hydroxylase 30S ribosomal protein S6 30S ribosomal protein S18 Chromosome partition protein Smc Alpha-ketoglutarate-dependent taurine dioxygenase tRNA threonylcarbamoyladenosine biosynthesis protein TsaE L(+)-tartrate dehydratase subunit alpha Undecaprenyl-diphosphatase Diguanylate cyclase YdeH Putative sensor-like histidine kinase YedV Putative transcriptional regulatory protein YedW Flap endonuclease Xni Putative acid--amine ligase YgiC 4,5-DOPA dioxygenase extradiol Inorganic triphosphatase Sensor histidine kinase YpdA Inner membrane protein YqiJ intégrité mb Inner membrane protein YqiK
**Genes encoding for virulence factors and stress response**	
Acyl carrier protein Cytoplasmic alpha-amylase 2,4-dienoyl-CoA reductase [NADPH] Modulator of FtsH protease HflK Respiratory nitrate reductase 2 alpha chain NADH-quinone oxidoreductase subunit G Epoxyqueuosine reductase Cytoplasmic trehalase Putative diguanylate cyclase YedQ	Multiple stress resistance protein BhsA Cytoskeleton bundling-enhancing protein CbeA Chaperone protein DnaJ Type II secretion system protein E N-acetyl-alpha-D-glucosaminyl-diphospho-ditrans, octacis-undecaprenol 4-epimerase Uroporphyrinogen decarboxylase Ascorbate-specific PTS system EIIC component Ascorbate-specific PTS system EIIA component Inner membrane ABC transporter ATP-binding protein YddA Membrane-bound lytic murein transglycosylase C Lipopolysaccharide export system permease protein LptF Lipopolysaccharide export system permease LptG Polysialic acid transport protein KpsD Serine protease sat autotransporter

* Similar gene shared by wild-type and target strains.

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
