# Peer review of "Escherichia coli Isolated from Diabetic Foot Osteomyelitis: Clonal Diversity, Resistance Profile, Virulence Potential, and Genome Adaptation"

_microorganisms, 2021, doi:10.3390/microorganisms9020380_

Round 1

Reviewer 1 Report

This is a very concise study, with a focus on the virulence potential of isolates obtained from osteomyelitis cases. I have only minor suggestions for revision.

lines 54-68: This paragraph is strangely written, and does not fit the style of the rest of the manuscript. The word "the" is missing in many places (eg. on chromosome, in human body, invade host, etc.). Please have this paragraph carefully checked for grammar.

line 75: Please provide a citation for the Helsinki Declaration.

line 114: Why was the Pasteur MLST scheme used? STs from the Achtman scheme are much more common, and things like ST131 are interesting for readers. Please consider changing or including Achtman STs throughout the entire manuscript.

line 148: An n= value would be helpful for the reader here.

line 150: Please list the library fragment size as well (eg. PE150 or PE250, etc.)

line 158: Additional programs could have been used here for AMR and virulence gene detection. The authors could use ARIBA with the Illumina reads directly, or with ABRicate on the assemblies. These tools would be very useful for finding genes that may have been missed with PCR.

line 182: 1.95 should be 1.94 instead.

line 220: Again, ARIBA and ABRicate might show additional genes.

line 226: The detected ciprofloxacin resistance is labelled "interesting" here, but is never mentioned in the discussion. It would be very useful to know what the resistance mechanisms are (eg. plasmid mediated qnr genes, gyrA and parC mutations?). This would be very useful information to have. A supplementary table of AMR genotype and phenotype information would also be useful.

line 302: A short discussion on the AMR findings would be appreciated here.

Author Response

This is a very concise study, with a focus on the virulence potential of isolates obtained from osteomyelitis cases. I have only minor suggestions for revision.

We thank the reviewer for these positive comments. We believe that we have improved the manuscript thanks to his/her comments.

lines 54-68: This paragraph is strangely written, and does not fit the style of the rest of the manuscript. The word "the" is missing in many places (eg. on chromosome, in human body, invade host, etc.). Please have this paragraph carefully checked for grammar.

We rewrote the paragraph in the new manuscript.

line 75: Please provide a citation for the Helsinki Declaration.

We added a reference (Ref. 15) in the new version of the manuscript.

line 114: Why was the Pasteur MLST scheme used? STs from the Achtman scheme are much more common, and things like ST131 are interesting for readers. Please consider changing or including Achtman STs throughout the entire manuscript.

We changed the results and only included the results of the Achtman MLST scheme (Figure 1).

line 148: An n= value would be helpful for the reader here.

We added this information in the new version of the manuscript.

line 150: Please list the library fragment size as well (eg. PE150 or PE250, etc.)

We added this information in the new version of the manuscript.

line 158: Additional programs could have been used here for AMR and virulence gene detection. The authors could use ARIBA with the Illumina reads directly, or with ABRicate on the assemblies. These tools would be very useful for finding genes that may have been missed with PCR.

Antimicrobial resistance genes were obtained from ABRicate and ResFinder softwares [Ref 28, 29] on assembled genomes. Only multidrug transporter A gene  (mdf(A)) was identified, coverage 100% for all isolates, identity 97.57% (NECR70, NECR107) and 98.14% (NECS21, NECS50). mdf(A) has a broad spectrum specificity that include, Erythromycin, Tetracycline, Rifampicin, Kanamycin, Chloramphenicol and Ciprofloxacin.

Strain

Gene

Coverage

Identity

Accession number

NECR70

mdf(A)

100%

97.57%

Y08743

NECR107

mdf(A)

100%

97.57%

Y08743

NECS21

mdf(A)

100%

98.14%

Y08743

NECS50

mdf(A)

100%

98.14%

Y08743

This information was added in the new version of the manuscript.

line 182: 1.95 should be 1.94 instead.

We modified accordingly.

line 220: Again, ARIBA and ABRicate might show additional genes.

We added additional information in the section 3.4 E. coli genome analysis.

line 226: The detected ciprofloxacin resistance is labelled "interesting" here, but is never mentioned in the discussion. It would be very useful to know what the resistance mechanisms are (eg. plasmid mediated qnr genes, gyrA and parC mutations?). This would be very useful information to have. A supplementary table of AMR genotype and phenotype information would also be useful.

The resistance mechanism of ciprofloxacin is exclusively due to gyrA and parC genes. No plasmid mediated quinolone resistances were found. We deleted “interestingly” because these mutations have been many times published.

line 302: A short discussion on the AMR findings would be appreciated here.

We gave informations in the new version of the manuscript : Ln 240-247 and Ln 368-372.

Reviewer 2 Report

Manuscript ID <microorganisms 1109414>

In this manuscript, a study on Escherichia coli strains isolated from diabetic foot infections and from diabetic foot osteomyelitis were carried out, comparing isolated strains based on several parameters: clonal diversity; resistence profile; virulence potential; genome adaptation.

This study characterised the E. coli strains isolated and demonstrated that these bacteria were highly genetically diverse with different pathogenicity traits.

Phylogenetic backgrounds, virulence factors and antibiotic resistance profiles were determined. Whole-genome analysis of E. coli strains isolated from same patients at different periods were performed.

The main part of the strains, was assigned to the virulent B2 and D phylogenetic groups, typically associated with more virulent strains. Concerning the genes encoding the adhesins, the fimbrial papG2 gene was significantly more detected in strains belonging to B2 phylogroup isolated from diabetic foot infections compared to diabetic foot osteomyelitis. The most prevalent antibiotic resistance pattern was as follow: ampicillin > cotrimoxazole > ciprofloxacin. Analysis of the genome of strains isolated at two periods in diabetic foot osteomyelitis showed a decrease of the genome size. A shared mutation on the putative acyl-CoA dehydrogenase-encoding gene aidB was observed on both strains. E. coli isolates from diabetic foot osteomyelitis were highly genetically diverse with different pathogenicity traits. An adaptation in the bone structure could require genome reduction and some important modifications in the balance virulence/resistance of the bacteria.

The novelty of this study deals with the determination for the first time the longitudinal evolution of E. coli genomes present at different times in bone. The virulome, the resistome and the housekeeping genes were modified. Thus, this suggests that E. coli strains modify their virulence by generating an adapted microbial population in the aim to survive in the bone and limit the host immune response.

In the Materials and methods sections it is reported that “A multiplex PCR was used for the detection of plasmidic blaampC genes. It would be interesting to have information about the plasmids behavior in these strains and their adaptation to bone colonization.

The manuscript is well written and add value and insights to studies of medical microbiology and it falls within aims and scope of the journal Microorganisms.

Revisions

Line 59: “… virulence factors (VFs) …” specify that it is referred to gene;

Line 125: “Total DNA …” specify “Total DNA of cultures…”

Line 127: “ESBLs” please, explain the acronym;

Table S1: “Other Enterococcus” change to Other Enterococcus spp.; “Coagulase-negative Streptococcus” change to ‘Coagulase-negative’ not in Italic style, as follows: “Coagulase-negative Streptococcus spp.”;

Table 3: “deep foot infections (dFI)”, instead of “diabetic foot infections (DFI)”, please specify and correct;

Lines 287-288: “in DFI (…) even if this difference was not significant (Table3).” Control, please, the difference between dFI and DFI;

Line 389: “E. coli” change to Italic style;

Line 563: “Pseudomonas aeruginosa” change to Italic style.

Author Response

In the Materials and methods sections it is reported that “A multiplex PCR was used for the detection of plasmidic blaampC genes. It would be interesting to have information about the plasmids behavior in these strains and their adaptation to bone colonization.

PlasmidFinder [Ref 30] was used for plasmids prediction. No plasmid was identified within the target genomes.

The manuscript is well written and add value and insights to studies of medical microbiology and it falls within aims and scope of the journal Microorganisms.

We thank the reviewer for these positive comments. We believe that we have improved the manuscript thanks to his/her comments.

Revisions

We modified accordingly along the manuscript the different errors:

Line 59: “… virulence factors (VFs) …” specify that it is referred to gene.

Line 125: “Total DNA …” specify “Total DNA of cultures…”

Line 127: “ESBLs” please, explain the acronym;

Table S1: “Other Enterococcus” change to Other Enterococcus spp.; “Coagulase-negative Streptococcus” change to ‘Coagulase-negative’ not in Italic style, as follows: “Coagulase-negative Streptococcus spp.”;

Table 3: “deep foot infections (dFI)”, instead of “diabetic foot infections (DFI)”, please specify and correct;

Line 389: “E. coli” change to Italic style

Line 563: “Pseudomonas aeruginosa” change to Italic style.

Lines 287-288: “in DFI (…) even if this difference was not significant (Table3).” Control, please, the difference between dFI and DFI

We controlled and corrected along the text.